# Collecting and Detecting Ancient Greek Historians through Wikibase and Wikidata

**Leonardo D'Addario**

**Affiliation:** Ancient History, Leipzig University

**Email:** leonardo.daddario@uni-leipzig.de

**ORCID:** 0009-0004-5271-3878

**Key words:** ancient greek historiography, polybius, wikibase, digital classics.

## 1. Introduction

This paper explores the potential of Wikibase and Wikidata as tools for cataloguing data and metadata on Ancient Greek historians. The study forms part of my PhD research in Ancient Greek History at Leipzig University, conducted under the supervision of Monica Berti (Leipzig University) and Stefan Schorn (KU Leuven) in the framework of MECANO,[1] a doctoral network funded by the Horizon Europe program for research and innovation, which investigates the dynamics of canonization in Graeco-Roman texts. In particular, the topic of my PhD focuses on *The Histories* of Polybius.

*The Histories* of Polybius (206–124 BCE, approx.) originally consisted of 40 books, covering the period from 264 BCE (the outbreak of the First Punic War) to 146 BCE (the destruction of Carthage and Corinth) and documenting the rise of Rome as the main political power in the Mediterranean world. Unfortunately, while only the first five books of his work are preserved, the others are known only through quotations in later sources or collections of excerpts created during the Byzantine Empire. Despite this lacuna, Polybius' work is fundamental for Classical studies, as it is the only surviving example of Hellenistic historiography and the primary source for many of the so-called "lost historians."[2]

The expression "lost historians" refers to ancient historiographers whose works are lost and are now known only through verbal quotations, paraphrases, and allusions in later sources. Following Schepens' (1997, 166-7 n. 66) influential term, I will call sources quoting earlier writers "cover-texts."[3] Passages containing citations are often extracted from the cover-texts, gathered into collections, and studied as "fragments" (i.e., surviving parts of a lost whole). This approach underpins one of the greatest philological works of the last century: Felix Jacoby's *Die Fragmente der Griechischen Historiker*, the authoritative collection of fragments of the lost historiographers. However, as Hau (2024, 151–52) emphasises, it is crucial, when dealing with fragments, to take into account their context of transmission, since one cannot be certain how much the cover-text altered the quoted originals or whether the quotation is verbatim or a paraphrase. Hence, the need to study the quoting and bibliographic practices of the cover-texts.

Given Polybius' status as a source for many lost historiographers, the project "Detecting and Retrieving Lost Historians" examines *The Histories* as a cover-text. The main goals are: (1) to analyse the language that Polybius uses when citing earlier writers in order to provide insights into

---

[1] https://mecano-dn.eu/

[2] According to the appendix of the Teubner edition, Polybius quotes fifty-two writers, among whom twenty-two are historiographers.

[3] Schepens (1997, 167n66) employs the term "cover-text" due to the triple meaning of the verb "to cover"— "to protect", "to conceal", and "to enclose." The cover-text functions in three interrelated ways: it preserves quotations of lost works (thereby "protecting" them from total oblivion); it conceals their original form, as most stylistic and contextual features of the quoted text are altered; and it encloses them within a new narrative framework, recontextualizing the quoted text to serve the aims of the quoting author.

his quoting practice and to assess his reliability as a source for fragments, and (2) to establish whether Polybius' quoting practice reflects engagement with a specific canon of authoritative writers.

The first step of the project involves cataloguing all passages where Polybius quotes other authors and collecting information about them. This catalogue must include:

- The original Greek text of the quoting passages.
- Information about the quoted authors (e.g., name, provenance, period) and their works (e.g., title, number of books, content).
- The relevant linguistic elements of the quoting text–namely, the Greek forms of the authors' names and their works, as well as verbs introducing citations.

Although the research focuses on lost historiographers, authors from other genres whose works are cited by Polybius will be included in the collection. The analysis of the final catalogue may indeed reveal that Polybius' quoting habit varies based on literary genres. Citations of preserved authors will also be included, as the comparison of the quoted text with its original may help evaluate Polybius' accuracy and determine whether he uses specific verbs or expressions to quote verbatim or paraphrase.

As MECANO fosters the integration of traditional approaches with new digital methods, I am currently developing a Wikibase instance called *The Library of Polybius*[4] in order to collect the quoting passages of *The Histories* and their metadata according to the principles stated above. In section 2 of this paper, I will discuss the data model I designed to describe each quoting passage through the statements system. In section 3, I will explore whether it is worth adding other and more specific statements to the data model.

While many digital practices are now being developed for studying the indirect transmission of classical authors and works,[5] Wikibase's application for this specific research field remains unexplored. However, Wikibase offers an ideal digital environment for cataloguing and analysing ancient Greek passages and their linguistic features.

Firstly, it allows for structuring data and metadata based on specific research questions. Through the statements system, one can describe each item by attributing to it the characteristics that need to be analysed. For instance, I designed a data model including statements specifying the relevant linguistic elements of each quotation, which are the very object of my research.

Secondly, the SPARQL Query Service allows for a fast and efficient analysis of data and metadata. In the specific case of *The Histories*, queries such as "quotations where Polybius uses the verb ἱστορέω" or "quotations where Polybius specifies the title of works" can help analyse Polybius' citing practice (or *ratio laudandi*, as classical scholars usually call it).

Lastly, Wikibase allows for the creation of datasets which will remain available and reusable for future researchers according to the principles of Linked Open Data (LOD).[6] In fact, cataloguing quoting passages is a typically philological task, and most scholars would do it manually and then present only the outcome of their analysis of the catalogue. However, such collections usually remain unpublished, meaning other scholars can only reuse the research results presented in an academic work.

## 2. Data model for quoting passages

The following table (*Figure* 1) shows the data model for quoting passages. This data model has been designed based on that of the Wikibase instance *Hypotheseis*, created by Camillo Carlo Pellizzari di

[4] https://lybrary-of-polybius.wikibase.cloud/wiki/Main_Page.
[5] Cf. Berti (202, 2024).
[6] On *Linked Open Data* (LOD) in Ancient World studies, see Cayless (2019).

San Gerolamo.[7] Each row represents the property-value pair. The first column specifies whether the pair is necessary or not. The central column includes the properties of each pair. The symbol [Q] means that the property-value pair is a qualifier. The symbol [R] means that the property-value pair is a reference. The third column includes the values of each pair. In the round brackets, the data type of each property is specified.

| Necessity | Proprieties | Values |
|---|---|---|
| yes | instance of | quoting passage (item) |
| yes | quoting text | Greek text (string) |
| yes | [Q] transcribed from the expression | expression (item) |
| yes | [R] CTS URN | CTS URN (external identifier) |
| yes | [R] URL of the transcription | URL to Perseus Scafe Viewer (url) |
| yes | taken from | work (item) |
| yes | [Q] citation | citation (string) |
| yes | quoted author(s) | person (item) |
| yes | [Q] textual form | Greek text (string) |
| yes (if "textual form" has a value) | [Q] lemma(s) | lemma (string) |
| yes (if "textual form" has a value) | [Q] reference form | proper name / toponym / generic noun (item) |
| yes | [R] determination method | explicit textual reference / inferred from the context (item) |
| yes (if "textual form" has a value) | [R] lemma reference(s) | *Logeion* URL (url) |
| yes | quoted work(s) | work (item) |
| yes (if there is the previous pair) | [Q] textual form | Greek text (string) |
| yes (if "textual form" has a value) | [Q] lemma(s) | lemma (string) |
| yes (if "textual form" has a value) | [Q] reference form | name / section (item) |
| yes | [R] determination method | explicit textual reference / inferred from the context (item) |
| yes (if "textual form" has a value) | [R] lemma reference(s) | *logeion* URL (url) |
| no | reporting verb(s) | lemma (string) |
| yes (if there is the statement) | [Q] textual form | Greek text (string) |
| yes (if there is the statement) | [Q] subject | person (item) |
| yes (if there is the statement) | [R] lemma reference(s) | *logeion* URL (url) |

*Figure 1*

In order to discuss the data model in detail, a concrete example–notably, Polyb. IV 20, 5 (Q6)–will be provided. In this passage, Polybius quotes the historian Ephorus and alludes to the content of the proem of his historiographical work: *οὐ γὰρ ἡγητέον μουσικήν, ὡς Ἔφορός φησιν ἐν τῷ προοιμίῳ τῆς ὅλης πραγματείας, οὐδαμῶς ἁρμόζοντα λόγον αὑτῷ ῥίψας, ἐπ᾽ ἀπάτῃ καὶ γοητείᾳ παρεισῆχθαι τοῖς ἀνθρώποις·* (Trans. For we must not suppose, as Ephorus, in the Preface to his History, making a hasty assertion quite unworthy of him, says, that music was introduced by men merely for the purpose of beguiling and bewitching.)[8] In the following paragraphs, I will show and explain the

[7] https://hypotheseis.wikibase.cloud/wiki/Pagina_principale.
[8] All translations of Polybius' text are from the Loeb edition.

statements describing the item, specifying their relevance to my research questions and providing guidelines for their correct use.

### 2.1 Quoting text, expression, and work

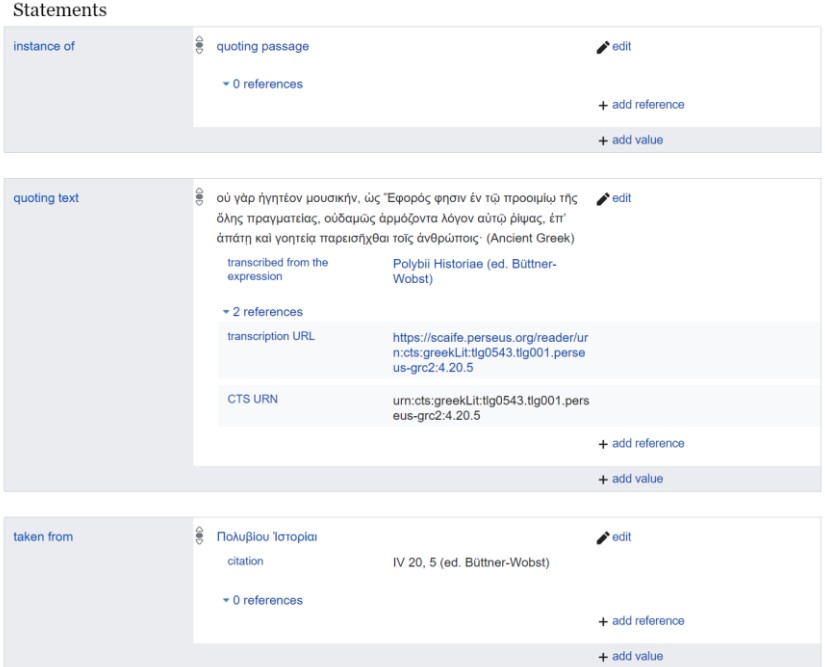

*Figure 2*

Let us take a look at *Figure* 2. The item Q6 is clearly an instance of a "quoting passage (Q3)", which is an entity. As shown in *Figure* 1, the data type of the property "quoting text (P7)" is a string. This property therefore has the ancient Greek text of the quoting passage as its value. The qualifier of the second statement specifies the critical edition from which the ancient Greek text has been transcribed. This is notably the Teubner edition by Büttner-Wobst, the XML file of which is openly available on the GitHub repository of the Perseus Digital Library.[9] The ancient Greek text has been copied from this XML file and pasted as the value of the property "transcribed from the expression (P8)". The first reference provides the URL to retrieve the exact passage of *The Histories* in the Perseus Digital Library Scaife Viewer, which is a digital reading environment for text collections. The second reference specifies the CTS URN of the quoting passage. (urn:cts:greekLit:tlg0543.tlg001.perseus-grc2:4.20.5), a standardized identifier for texts in digital environments designed to uniquely and persistently reference specific passages.[10]

Thus, in the property "transcribed from the expression (P8)", *expression* denotes Polybius' Greek text as edited by Büttner-Wobst. The term *expression* is employed according to the IFLA LRM model. This conceptual entity-relationship model specifically distinguishes *expression* and *work* as different entities.[11] *Work* is defined as an "intellectual or artistic content of a distinct creation"–in our case, the abstract and intangible idea of *The Histories*, which exists independently of language, form, or physical format. On the other hand, *expression* is defined as "a distinct combination of signs conveying intellectual and artistic content"–here, the specific version of Polybius' text as edited by

---

[9] https://github.com/PerseusDL/canonical-greekLit/tree/master/data/tlg0543/tlg001.

[10] On the use of CTS URNs for citing and retrieving Classical sources, see Babeu (2019), and Backwell, Christopher, and Smith (2019). The URN "urn:cts:greekLit:tlg0543.tlg001.perseus-grc2:4.20.5" follows the Canonical Text Services standard ("cts"), where "tlg0543" identifies Polybius, "tlg001" refers to *The Histories*, "perseus-grc2" denotes the specific edition used–the Perseus XML file of the Büttner-Wobst's Teubner edition, and 4.20.5 references Book 4, Chapter 20, Section 5.

[11] I use the italic when I refer to *work* and *expression* as entities of the IFLA LRM model.

the Teubner editors.[12] Consequently, while the item "Polybii Historiae (ed. Büttner-Wobst) (Q7)" is an instance of *expression*, the item "Πολυβίου Ἱστορίαι (Q8)", the value of the third statement, is an instance of *work*. The original ancient Greek title is used here to preserve the historical identity of the *work*, as Polybius conceived the idea of *The Histories* in ancient Greek.[13] Additionally, the original Greek title ensures disambiguation, as translated titles are assigned to *expressions* according to the cataloguing standards of many libraries (e.g., LoC and VIAF).

The third statement ("taken from – Πολυβίου Ἱστορίαι") specifies the source of the quoting passage. The property "taken from (P11)" has the *work* as its value, because the quoting passage is a part of the intellectual and artistic content of *The Histories*. The statement also includes a qualifier specifying in which section of *The Histories* the quoting passage is found. The qualifier's value, "IV 20, 5 (ed. Büttner-Wobst)," refers to the *expression*. As a matter of fact, although the book division of *The Histories* could be considered as a characteristic of the *work* (since Polybius himself conceived it), the paragraph and subparagraph numbering belongs to the expression—specifically, the Teubner edition's textual configuration[14].

Before moving to the next section, a clarification about the criteria for selecting the quoting passage is required. In general, a quoting passage is defined as a section of *The Histories* where Polybius refers to an author and their work. This implies that all passages naming a specific author but not referring to the content of their work (e.g., anecdote about the author's life) will be excluded from the Wikibase. Fortunately, this almost never happens in Polybius, who cites other authors just to express his judgement about their literary and historical value. However, there are some exceptions, such as Aratus of Sicyon. He wrote the Ὑπομνήματα, a fundamental source for Polybius. At the same time, he was also a politician, and this is why Polybius mentions him multiple times. Therefore, *The Library of Polybius* will include only passages referring to Aratus of Sicyon as the writer of Ὑπομνήματα.

Then, since Wikibase allows for a maximum length of four hundred characters for a string, additional principles are strictly observed to delimit the quoting passage. Notably, the quoting passage will consist of a meaningful sentence and stop at the period or high dot. It is not important if Polybius continues describing the content of the lost work after the punctuation, as my research focuses exclusively on his quoting language–specifically, authors' names, works' titles, and reporting verbs. If the next sentence contains any of these three linguistic elements, it will be considered as another quoting passage. Consider Polyb. II 56, 6-7:

| | |
|---|---|
| [6] *βουλόμενος δὴ διασαφεῖν τὴν ὠμότητα τὴν Ἀντιγόνου καὶ Μακεδόνων, ἅμα δὲ τούτοις τὴν Ἀράτου καὶ τῶν Ἀχαιῶν, φησὶ τοὺς Μαντινέας γενομένους ὑποχειρίους μεγάλοις περιπεσεῖν ἀτυχήμασι, καὶ τὴν ἀρχαιοτάτην καὶ μεγίστην πόλιν τῶν κατὰ τὴν Ἀρκαδίαν τηλικαύταις παλαῖσαι συμφοραῖς ὥστε πάντας εἰς ἐπίστασιν καὶ δάκρυα τοὺς Ἕλληνας ἀγαγεῖν. [7] σπουδάζων δ' εἰς ἔλεον ἐκκαλεῖσθαι τοὺς ἀναγινώσκοντας καὶ συμπαθεῖς ποιεῖν τοῖς λεγομένοις, εἰσάγει περιπλοκὰς γυναικῶν καὶ κόμας διερριμμένας καὶ μαστῶν ἐκβολάς, πρὸς δὲ τούτοις δάκρυα καὶ θρήνους ἀνδρῶν καὶ γυναικῶν ἀναμὶξ τέκνοις καὶ γονεῦσι γηραιοῖς ἀπαγομένων.* | [6] Wishing, for instance, to insist on the cruelty of Antigonus and the Macedonians and also on that of Aratus and the Achaeans, he tells us that the Mantineans, when they surrendered, were exposed to terrible sufferings and that such were the misfortunes that overtook this, the most ancient and greatest city in Arcadia, as to impress deeply and move to tears all the Greeks. [7] In his eagerness to arouse the pity and attention of his readers he treats us to a picture of clinging women with their hair disheveled and their breasts bare, or again of crowds of both sexes together with their children and aged parents weeping and lamenting as they are led away to slavery. |

[12] Cf. Riva, Le Boeuf, and Žumer (2017, 21-2).

[13] Because the original titles of lost works were not fixed, the ancient Greek titles used in this study align with those assigned by Jacoby in *Die Fragmente der Griechischen Historiker*. For authors who are not historians, the ancient Greek titles follow the specific edition of reference. Since works by different authors may share the same title (e.g., Ἱστορίαι), the genitive form of the author's name is occasionally used for disambiguation.

[14] The Teubner edition (Büttner-Wobst 1882–1904) inherited this paragraph/subparagraph division from Bekker's earlier edition (1844). However, as the structural markup is materially embedded in the Teubner text, it remains a property of that *expression* within the IFLA LRM model.

According to the principles stated above, subparagraph 6 and subparagraph 7 will constitute two different quoting passages and therefore two different Wikibase items, respectively Q43 and Q44.

### 2.2 Quoted author(s) and quoted work(s)

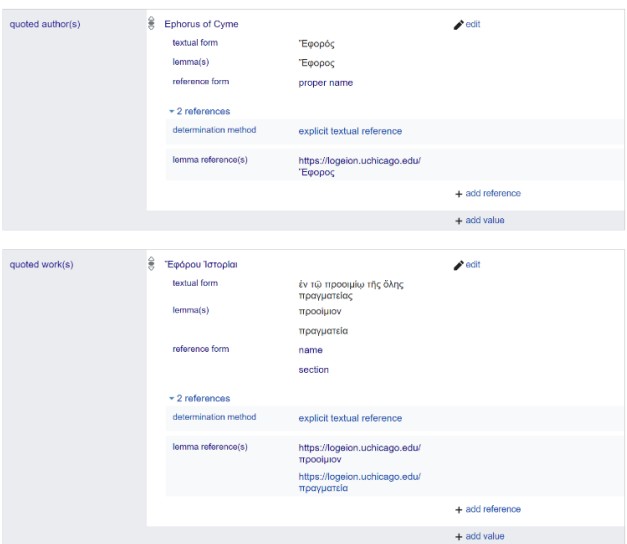

*Figure 3*

The statements in *Figure* 3 identify the quoted author and work and describe metadata about the quotation–notably, its linguistic characteristics. The value of the property "quoted author(s) (P13)" is the item "Ephorus of Cyme (Q9)", which is an instance of person. On the other hand, the value of "quoted work(s) (P15)" is the item "Ἐφόρου Ἱστορίαι (Q13)", which is an instance of *work*.

Quoted works, such as Ephorus' *Histories*, are always described as *works* because the *expressions* Polybius drew upon are unknown. Paraphrases or allusions to quoted works are indeed *expressions* of the cover-text's *work*, since they do not preserve the original combination of signs conveying the intellectual content. This principle extends to verbatim quotations as well: even when Polybius cites the text directly, he incorporates the quotation into the *expression* of his *work*, potentially altering the original wording or context of Ephorus.

The first qualifier of both statements highlights the exact textual form of the quotations. The value of the property "textual form (P17)" is a string representing Polybius' words referring to the quoted author and work (Ἔφορός and ἐν τῷ προοιμίῳ τῆς ὅλης πραγματείας). The second qualifier specifies the lemma of Polybius' quoting words. In the statement about the quoted author, the lemma is Ἔφορος. The statement about the quoted work has instead two lemmas, as Polybius' text refers both to Ephorus' work (πραγματείας) and to a specific section of it (προοιμίῳ). The last qualifier describes the quotation form. As shown in *Figure* 1, the property "reference form (P18)" is the same, but the items constituting its possible values change based on the statement.

As for "quoted work(s)", *Figure* 1 shows that the possible values of the "reference form (P18)" are the items "name (Q14)" and "section (Q18)". In Q6, both items are values of P18. However, there are examples where only one of these items is required. Consider "Polyb. X 44, 1 (Q17)": *Αἰνείας δὲ βουληθεὶς διορθώσασθαι τὴν τοιαύτην ἀπορίαν, ὁ τὰ περὶ τῶν Στρατηγικῶν ὑπομνήματα συντεταγμένος, βραχὺ μέν τι προεβίβασε, τοῦ γε μὴν δέοντος ἀκμὴν πάμπολυ τὸ κατὰ τὴν ἐπίνοιαν ἀπελείφθη* (Trans. Aeneas, the author of the work on strategy, wishing to find a remedy for the difficulty, advanced matters a little, but his device still fell far short of our requirements, as can be

seen from this description of it.) Here, Polybius only refers to Aeneas' work (τὰ περὶ τῶν Στρατηγικῶν ὑπομνήματα); the qualifier thus requires only "name"[15] as its value.

As for "quoted author(s)", the possible values of P18 are "proper name (Q15)", "toponym (Q20)", and "generic noun (Q21)". In Q6, Polybius refers to Ephorus only by his proper name, meaning that P18 has only the "proper name (Q15)" as its value. In the item Q22, P18 will instead have both the values "proper names (Q15)" and "toponym (Q20)", as Polybius specifies the provenance of the quoted historian: *ταῦτα δ' ἔστι συνεχῆ τοῖς τελευταίοις τῆς παρ' (Ἀρ)άτου Σικυωνίου συντάξεως* (Trans. These events immediately succeed those related at the end of the work of Aratus of Sicyon.) The item "generic noun (Q21)" will be used when the quoted authors are identified by a noun indicating the literary genre of their work, such as in Q27: *οὐ γὰρ ἂν Ἀρχέδικος ὁ κωμῳδιογράφος ἔλεγε ταῦτα μόνος* (Trans. For in that case not only Archedicus, the comic poet, would... have said this). Here, Polybius refers to Archedicus both by his proper name (Ἀρχέδικος) and by a noun indicating his role as comedy writer (κωμῳδιογράφος).

Both statements have two references. The data model (*Figure* 1) shows that the property "determination method (P14)" has two items as possible values: "explicit textual reference (Q12)", and "inferred from the context (Q33)". This reference is necessary to indicate how the Wikibase editor can verify that the quoted authors and works match those declared in the statements. As a matter of fact, while many quoting passages name the author and the work, there are cases where this must be inferred from the context. Consider "Polyb. XVI 19, 1 (Q34)": *Μετὰ δὲ ταῦτά φησι καταπροτερουμένην τὴν φάλαγγα ταῖς εὐχειρίαις καὶ πιεζομένην ὑπὸ τῶν Αἰτωλῶν ἀναχωρεῖν ἐπὶ πόδα, τὰ (δὲ) θηρία τοὺς ἐγκλίνοντας ἐκδεχόμενα καὶ συμπίπτοντα τοῖς πολμίοις μεγάλην παρέχεσθαι χρείαν* (Transl. Next he states that the phalanx, proving inferior in fighting power and pressed hard by the Aetolians, retreated slowly, but that the elephants were of great service in receiving them in their retreat and engaging the enemy.) Here, Polybius does not specify neither the author's name nor his work's. Let us take a look at *Figure* 4 showing the statements about the author and work of Q34.

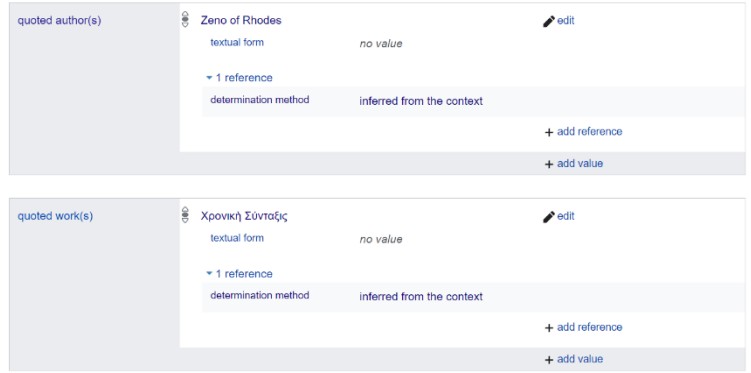

*Figure 4*

The broader context of the passages clarifies that Polybius is quoting the work of Zeno of Rhodes. The Wikibase editor knows that and must specify it in the reference.[16] It is worth noting that in such a case, the editor must use the statements about the quoted authors and works, but the qualifier indicating the textual form of the quotations will clearly have "no value".

---

[15] Since Polybius refers to the same work using different nouns—which cannot always be considered titles—the item has been labeled as "name" rather than "title."

[16] The item "inferred from the context (Q33)" is also used when only the section of the work only is specified (e.g., Q40), and when the author is identified only through a toponym, or generic noun (e.g., Q37).

Returning to *Figure* 3, the final references include URLs linking to the lemma entries in *Logeion*, an open access database of Latin and Ancient Greek dictionaries.[17]

### 2.3 Reporting verb(s)

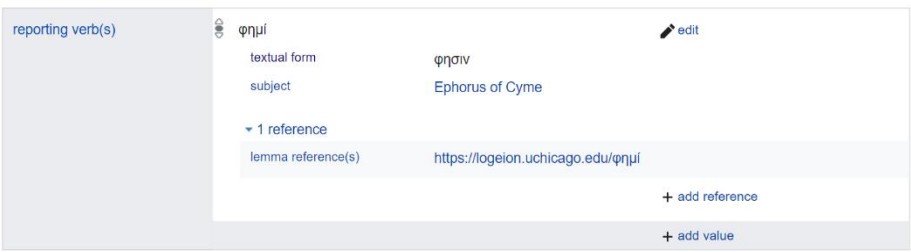

*Figure 5*

*Figure* 5 shows the final statement of Q6, specifying the "reporting verbs". These are verbs somewhat associated with the content of the quoted works, such as those denoting the act of saying, writing, or composing. The value of the property "reporting verb(s) (P24)" is a string representing the lemma of the verb. The statement includes two qualifiers. While the first specifies the verb's form as it appears in Polybius' text, the second identifies the verb's subject. The subject qualifier is essential for quoting passages where multiple verbs are attributed to multiple authors. Consider "Polyb. XII 13, 7 (Q27)," where λέγω has Archedicus as its subject, and φημί refers to Timaeus: *οὐ γὰρ ἂν Ἀρχέδικος ὁ κωμῳδιογράφος ἔλεγε ταῦτα μόνος περὶ Δημοχάρους, ὡς Τίμαιός φησιν* (Trans. For in that case not only Archedicus, the comic poet, would, as Timaeus asserts, have said this). Once the Wikibase is completed, this data structure will enable users to investigate, through the SPARQL Query Service, the verbs Polybius uses for different authors.

If we look at *Figure* 1 again, we will notice that the statement about the reporting verbs is not necessary. This is why there are also quoting passages where Polybius does not use a reporting verb, such as "Polyb. I 3, 2 (Q22)," cited above: *ταῦτα δ' ἔστι συνεχῆ τοῖς τελευταίοις τῆς παρ' (Ἀρ)άτου Σικυωνίου συντάξεως* (Trans. These events immediately succeed those related at the end of the work of Aratus of Sicyon.) In such cases, the Wikibase editor must omit the statement entirely, since there is no verb to analyse.[18] The special item "no value" must not be used here, as it implies that the verb should be there, but is missing.

### 3. The quotation type: a possible model expansion?

Since *The Library of Polybius* is still an ongoing project, additional statements are now being considered for inclusion in the data model. In this section, I will briefly discuss one of them and highlight its issues in order to clarify why they have not been added to the model.

When dealing with historical fragments, classical philologists are accustomed to distinguishing between four quotation types: verbatim quotations, paraphrases, allusions, and epitomizations.[19] While the term "verbatim quotations" is self-explanatory, the others require an explanation. Let us begin with paraphrases. A cover-text paraphrases the quoted work when it preserves the original words but adapts the syntax to the new context. This happens, for instance, when a quoted text becomes a subordinate clause in the cover-text. On the other hand, allusions are usually identified as brief and general references to the content of the quoted work, without reporting the specific wording of the author. Allusions often take the form of parenthetical clauses, such as: "This fact

---

[17] https://logeion.uchicago.edu/ λόγος.
[18] In passages like these, it may be worth introducing a property-value pair to highlight the use of παρά or similar prepositions. This approach could be extended to both "quoted author(s)" and "quoted work(s)" statements. For example, in Q6, such a property-value pair could indicate that Polybius uses ἐν when quoting Ephorus' *Histories*. This possibility is currently under consideration.
[19] Cf. Brunt (1980).

happened in Athens, as Timaeus also says". But their form may also vary. Consider Polyb. I 3, 2, cited above: *ταῦτα δ' ἔστι συνεχῆ τοῖς τελευταίοις τῆς παρ' (Ἀρ)άτου Σικυωνίου συντάξεως* (Trans. These events immediately succeed those related at the end of the work of Aratus of Sicyon.) Here, Polybius is simply alluding to the content of the *Memoirs* of Aratus of Sicyon. Epitomizations are instead defined as brief summaries of the content of the quoted work.

Based on these principles, one might consider creating a property "quotation type" with an item as data type. The possible value for this statement would then be four items corresponding to the four quotation types described above. However, while determining the quotation type is sometimes straightforward, most quoting passages are not easily classified. Consider the following quoting passage, "Polyb. II 56, 6 (Q43)", where Polybius quotes Phylarchus:

| | |
|---|---|
| *βουλόμενος δὴ διασαφεῖν τὴν ὠμότητα τὴν Ἀντιγόνου καὶ Μακεδόνων, ἅμα δὲ τούτοις τὴν Ἀράτου καὶ τῶν Ἀχαιῶν, φησὶ τοὺς Μαντινέας γενομένους ὑποχειρίους μεγάλοις περιπεσεῖν ἀτυχήμασι, καὶ τὴν ἀρχαιοτάτην καὶ μεγίστην πόλιν τῶν κατὰ τὴν Ἀρκαδίαν τηλικαύταις παλαῖσαι συμφοραῖς ὥστε πάντας εἰς ἐπίστασιν καὶ δάκρυα τοὺς Ἕλληνας ἀγαγεῖν.* | Wishing, for instance, to insist on the cruelty of Antigonus and the Macedonians and also on that of Aratus and the Achaeans, he tells us that the Mantineans, when they surrendered, were exposed to terrible sufferings and that such were the misfortunes that overtook this, the most ancient and greatest city in Arcadia, as to impress deeply and move to tears all the Greeks. |

Landucci (2017) maintains that the passage does not preserve Phylarchus' original words. While she does not explicitly state whether she considers it an epitomization, she certainly implies this. On the other hand, Kurpios (2020) argues that Polybius preserves some of Phylarchus' *ipsissima verba* here, suggesting the quoted passage represents a form of paraphrase. Adding a statement about the quotation type to this passage would create more problems than it solves. First, which scholar should the Wikibase editor follow? Should the editor express their own judgement on each such passage? This would require excessive time. Moreover, scholarly practice demonstrates that the categories mentioned above are often too rigid. Neither Landucci nor Kurpios explicitly specifies whether they consider the passage an epitomization or a paraphrase, but rather allude to these concepts generally. Expressing a definite judgement on this passage would indeed be very difficult. For these reasons, this statement has not been included in the data model.

## 4. Conclusion

The main objective of *The Library of Polybius* is to adapt a traditional philological approach to a digital environment in order to demonstrate how new computational methods can enhance classical scholarly approaches. The Wikibase is still under development; however, the paper has demonstrated its effectiveness in systematically cataloguing data and metadata from Polybius' quoting passages. The statements not only specify which authors and works Polybius quotes, but also highlight elements of his quoting language and briefly describe them. The project is still ongoing; therefore, there is room for further development. As more items are added to the Wikibase, the data model will evolve to accommodate new and more specific quoting passages. For the time being, the areas for further development are (1) how to highlight other linguistic elements that may prove relevant for this research (e.g., prepositions introducing complements), and (2) automating the integration of Wikidata's entries about ancient authors. Wikidata is indeed a very rich database, with many well-curated entries about the ancient world, such as those for Ephorus and his *Histories*.[20] Therefore, integrating them into the Wikibase rather than simply creating a Wikidata item property would be highly beneficial and would save considerable time for the Wikibase editor.

---

[20] Ephorus Wikidata item: https://www.wikidata.org/wiki/Q313791. Ephorus' *Histories* Wikidata item: https://www.wikidata.org/wiki/Q42190991.

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
