# OpenReview forum: "Collecting and Detecting Ancient Greek Historians through Wikibase and Wikidata"
_wikimedia.it/Wikidata_and_Research/2025/Conference — WD&R Paper_

### Official Review · ~Lucia_Sardo1 · 2025-01-07
**revisione**

**Originality:** 5
**Impact:** 5
**Confidence:** 4

**Review:**

La proposta risulta ben strutturata, con una chiara esposizione della metodologia usata, degli step del progetto e dei possibili risultati soprattutto in termini di impatto per la comunità scientifica di riferimento.

**Compliance:**

5

**Scientific Quality:**

5

---

### Official Review · ~Carlo_Bianchini1 · 2025-01-09
**LOD datasets for Polibius and lost ancient Greek historians**

**Originality:** 5
**Impact:** 4
**Confidence:** 4

**Review:**

The project aims to fill the present gap of lack of coherently structured data and metadata about ancient authors and their texts through both the creation and collection of structured data and the discover of relevant pattern within them. As the submission is focused on the presentation of an up-to-come project, I would suggest to put the proposal to the paper section, instead of the paper section.

**Compliance:**

5

**Scientific Quality:**

4

---

### Decision · Program_Chairs · 2025-02-05

Accept (Paper)